

# A double-decomposition based parallel exact algorithm for the feedback length minimization problem

Zhen Shang[1], Jin-Kao Hao[2] and Fei Ma[1]

[1] School of Economics and Management, Chang'an University, Xi'an, China
[2] LERIA, Université d'Angers, Angers, France

## ABSTRACT

Product development projects usually contain many interrelated activities with complex information dependences, which induce activity rework, project delay and cost overrun. To reduce negative impacts, scheduling interrelated activities in an appropriate sequence is an important issue for project managers. This study develops a double-decomposition based parallel branch-and-prune algorithm, to determine the optimal activity sequence that minimizes the total feedback length (FLMP). This algorithm decomposes FLMP from two perspectives, which enables the use of all available computing resources to solve subproblems concurrently. In addition, we propose a result-compression strategy and a hash-address strategy to enhance this algorithm. Experimental results indicate that our algorithm can find the optimal sequence for FLMP up to 27 activities within 1 h, and outperforms state of the art exact algorithms.

## INTRODUCTION

Enterprises face more and more competition, which requires the competitors to develop new products in a short time. However, product development projects often involve many interrelated activities with complex information dependences (*Lin et al., 2012*; *Bashir et al., 2022*). Such activities usually follow uncertain processes and rework frequently, which makes it difficult for managers to control the project durations, costs and risks (*Mohammadi, Sajadi & Tavakoli, 2014*; *Lin et al., 2018*). Therefore, how to sequence interrelated activities to reduce negative impacts has drawn considerable attention (*Attari-Shendi, Saidi-Mehrabad & Gheidar-Kheljani, 2019*; *Wen et al., 2021*).

The design structure matrix (DSM) can clearly describe interrelated activities and interdependence, which is considered an effective tool in scheduling development projects (*Browning, 2015*; *Wen et al., 2021*). Figure 1A presents a typical DSM of a balancing machine project (*Abdelsalam & Bao, 2007*), where activities are listed on the left column and the top row following the same order; $d_{i,j}(0 \leq d_{i,j} \leq 1, i \neq j)$ denotes the degree of information dependence of activity $i$ on $j$(marked in red). Since activity $i$ precedes $j$, $d_{i,j}$ represents the backward information flow from downstream to upstream in the activity

Corresponding author
Fei Ma, mafeixa@chd.edu.cn

| Activities | | 1 | 2 | ... | i | ... | j | ... | n |
|---|---|---|---|---|---|---|---|---|---|
| Simulation analysis motion and cycle time | 1 | | 0.1 | | | | $d_{1,j}$ | | $d_{1,n}$ |
| Rotary drive component design | 2 | | | | | | | | $d_{2,n}$ |
| ... | ... | | | | | | | | |
| Lift drive component design | i | | $d_{i,2}$ | | | | $d_{i,j}$ | | $d_{i,n}$ |
| ... | ... | | | | | | | | |
| Optimal component design of correcting indexing unit | j | $d_{j,1}$ | | | $d_{j,i}$ | | | | |
| ... | ... | | | | | | | | |
| Component design of high intensity correction spindle | n | | $d_{n,2}$ | | $d_{n,i}$ | | $d_{n,j}$ | | |

(a)

| Activities | | 1 | 2 | ... | j | ... | i | ... | n |
|---|---|---|---|---|---|---|---|---|---|
| Simulation analysis motion and cycle time | 1 | | 0.1 | | $d_{1,j}$ | | | | $d_{1,n}$ |
| Rotary drive component design | 2 | | | | | | | | $d_{2,n}$ |
| ... | ... | | | | | Feedback | | | |
| Optimal component design of correcting indexing unit | j | $d_{j,1}$ | | | | | $d_{j,i}$ | | |
| ... | ... | | | | | | | | |
| Lift drive component design | i | | $d_{i,2}$ | | $d_{i,j}$ | | | | $d_{i,n}$ |
| ... | ... | | | Feedforward | | | | | |
| Component design of high intensity correction spindle | n | | $d_{n,2}$ | | $d_{n,j}$ | | $d_{n,i}$ | | |

(b)

**Figure 1** Illustrations of DSM.

sequence, which is above the diagonal and called feedback; $d_{j,i}$ is the information flow in opposite direction, which is under the diagonal and called feedforward. In Fig. 1B, if the order of activities $i$ and $j$ is reversed, then $d_{i,j}$ and $d_{j,i}$ become a feedforward and a feedback, respectively. The information flows from other activities to $i$ and $j$ are also affected (marked in yellow), which means that adjusting the activity sequence can significantly affect the overall information flows in DSM (*Lin et al., 2015*; *Meier et al., 2016*).

Figure 1 indicates that due to the existence of feedbacks, upstream activities often execute in the absence of information. Once the downstream activities complete, feedbacks may cause upstream activities to rework. In fact, feedbacks usually involve suggestions, errors and modifications, which are the main reason for project delay and cost overrun (*Haller et al., 2015*; *Lin et al., 2015*; *Wynn & Eckert, 2017*). Therefore, some researches suggest minimizing the total feedback values of activity sequence to reduce negative effects (*Qian et al., 2011*; *Nonsiri et al., 2014*). However, these studies do not consider the influence of feedback length, *i.e.,* long feedbacks across more activities may cause more upstream activities to rework than short ones. Hence, the objective of minimizing the total feedback length is proposed, which has been widely applied in DSM-based scheduling problems. For

instance, *Qian & Yang (2014)* demonstrated the effectiveness of optimizing the feedback length to reduce the overall reworks through a case study of a pressure reducer project. *Benkhider & Kherbachi (2020)* used a composite objective that considers the feedback length to reduce the duration of Huawei P30 pro project; *Gheidar-kheljani (2022)* studied a two-objectives scheduling model that considers the feedback length and the cost of decreasing dependence among activities.

For a development project, let $I = \{1, 2, \ldots, i, \ldots, n\}$ be an activity set, $D = (d_{i,j})_{(n \times n)}$ be a DSM, and decision variable $x_{i,j} = 1$ if activity $i$ precedes $j$, otherwise $x_{i,j} = 0$. Then, the feedback length minimization problem (FLMP) can be formulated as a 0-1 quadratic programming problem (*Qian & Lin, 2013*):

$$Min \sum_{i=1}^{n} \sum_{j=1, j \neq i}^{n} d_{i,j} x_{i,j} \left( \sum_{k=1, k \neq j}^{n} x_{k,j} - \sum_{k=1, k \neq i}^{n} x_{k,i} \right) \tag{1}$$

$$s.t. \, x_{i,j} + x_{j,i} = 1, for \, 1 \leq i < j \leq n \tag{2}$$

$$x_{i,j} + x_{j,k} + x_{k,i} \leq 2, for \, i \neq j \neq k \tag{3}$$

$$x_{i,j} \in \{0, 1\}, for \, 1 \leq i, j \leq n, i \neq j. \tag{4}$$

where 0–1 vector $X = (x_{1,2}, \ldots, x_{i,j}, \ldots, x_{n,n-1})$ denotes an activity sequence; objective function Eq. (1) minimizes the total feedback length, if $x_{i,j} = 1$, then feedback $d_{i,j}$ and its length $(\sum_{k=1, k \neq j}^{n} x_{k,j} - \sum_{k=1, k \neq i}^{n} x_{k,i})$ are counted into the objective value; constraint Eq. (2) guarantees that there is only one execution order for activity $i$ and $j$; constraint Eq. (3) ensures that the execution order is transitive; constraint Eq. (4) guarantees that decision variables are binary.

Further, the original model can be simplified to a sequence-based model (*Lancaster & Cheng, 2008*; *Shang et al., 2019*). Let integer vector $S = (s_1, s_2, \ldots, s_h, \ldots, s_k, \ldots, s_n)$ be an activity sequence, where decision variable $s_h$ is the activity at position $h$ of the sequence, for example, $s_3 = 5$ means that activity 5 is assigned to position 3. Since position $h$ is set before position $k(h < k)$, $d_{s_h, s_k}$ is the feedback from position $k$ to $h$, and the sequence-based model can be formulated as follows:

$$Min \sum_{h=1}^{n-1} \sum_{k=h+1}^{n} d_{s_h, s_k} (k - h) \tag{5}$$

$$s.t. \, s_h, s_k \in I, s_h \neq s_k, d_{s_h, s_k} \in D, for \, 1 \leq h < k \leq n \tag{6}$$

where objective function Eq. (5) minimizes the total feedback length, $(k - h)$ is the length of feedback $d_{s_h, s_k}$; constraint Eq. (6) limits the values of the decision variables $(s_h, s_k \in I)$ and prohibits an activity from appearing in multiple positions $(s_h \neq s_k)$. Due to the concise expression of FLMP, we mainly analyze the sequence-based model in the rest of this article.

Researchers have proved that FLMP is NP-hard and extremely difficult to solve (*Meier, Yassine & Browning, 2007*). Therefore, many studies proposed heuristic approaches to obtain near-optimal activity sequences. These algorithms usually follow the classic heuristic framework, such as genetic algorithm, local search, which can obtain a reasonable solution within a short time, but cannot guarantee the global optimum. On the other hand, studies on exact approach are quite limited, and the existing algorithms are not practical due to the weak computational capability. Nevertheless, the research on specialized exact algorithms has promoted the exploration FLMP properties. *Shang et al. (2019)* found that FLMP has optimal sub-structure, which allows the original problem to be decomposed into multiple subproblems. Based on this property, they developed a parallel exact algorithm, which can solve FLMP with 25 activities in 1 h and is the state of the art exact approach in current literature (see the detailed review in 'Literature review').

This study focus on improving the computational capability of exact approach for FLMP, through fully utilizing the structural properties. We develop a double-decomposition based parallel branch-and-prune algorithm (DDPBP), to obtain the optimal activity sequence. The proposed algorithm first divides FLMP into forward and backward scheduling subproblems, then decomposes subproblems into several scheduling tasks and solving them concurrently. The resulting optimal subsequences are connected to be the global optimum. Furthermore, we propose an effective result-compression strategy to reduce communication costs in parallel process, and a novel hash-address strategy to boost the efficiency of sequence comparisons. Computational experiments on 480 FLMP instances show that DDPBP significantly reduces the time consumption for obtaining the optimal solution, and increases the problem scale that exact algorithms can solve to 27 activities within 1 h.

The rest of this article is organized as follows. 'Literature review' presents a literature review on exact and heuristic approaches for FLMP. In 'FLMP analysis', we recall the properties of FLMP, which is the foundation of the proposed algorithm. 'Double-decomposition based algorithm' introduces the main scheme and key phases of DDPBP, including the result-compression strategy. 'Hash strategy' provides the hash-address strategy applied in DDPBP. 'Computational experiments' conducts the comparisons between DDPBP and state of the art algorithms. 'Analysis' provides systematic analyses of parameters and key strategies. 'Conclusions' draws conclusions.

## LITERATURE REVIEW

This section presents a literature review about FLMP and the existing solution approaches, while some closely related problems are also mentioned. Table 1 summarizes the optimization objectives and the proposed algorithms discussed in the literature.

The high uncertainty of the development process makes it difficult to estimate the project duration, costs and risks. Therefore, many studies usually introduce alternative

**Table 1  Literature summary.**

| Representative literature | Optimization objectives | Proposed algorithm |
|---|---|---|
| Exact approach | | |
| *Qian & Lin (2013)* | Total feedback length minimization | CPLEX MILP solver |
| *Shang et al. (2019)* | Total feedback length minimization | A hash-address based parallel branch-and-prune algorithm |
| *Gheidar-kheljani (2022)* | Multi-objective: feedback length, cost of decreasing dependence | CPLEX solver for small and medium problems |
| Heuristic approach | | |
| *Altus, Kroo & Gage (1995)* | Total feedback length minimization | A genetic algorithm |
| *Todd (1997)* | Total feedback length minimization | A multiple criteria genetic algorithm |
| *Meier, Yassine & Browning (2007)* | Total feedback length minimization | A genetic algorithm |
| *Lancaster & Cheng (2008)* | Total feedback length minimization | A parameter adaptive evolutionary algorithm |
| *Qian et al. (2011)* | Total feedback value minimization | A hybrid Algorithm based on local search and LIP-solver |
| *Qian & Yang (2014)* | Total feedback length minimization | An exchange-based local search heuristic |
| *Lin et al. (2015)* | Total feedback time minimization | A hybrid Algorithm based on local search and BLP-solver |
| *Lin et al. (2018)* | Total feedback length minimization | A hybrid Algorithm based on insertion-based heuristic and simulated annealing |
| *Attari-Shendi, Saidi-Mehrabad & Gheidar-Kheljani (2019)* | Multi-objective: feedback value, technology risk and financial status | A fuzzy interactive method |
| *Khanmirza, Haghbeigi & Yazdanjue (2021)* | Total feedback length minimization | An enhanced imperialist competitive algorithm |
| *Wen et al. (2021)* | First and second order rework time minimization | An insertion-based heuristic algorithm |
| *Peykani et al. (2023)* | Multi-objective:feedback length, project duration | A hybrid approach based on genetic algorithm |

optimization objectives to reschedule the development process. One fundamental objective is to minimize the total feedback value, *i.e.*, the overall strength of the information flows above the diagonal of DSM. *Qian et al. (2011)* simplified this scheduling problem by treating a group of activities as one abstract activity, then developed a hybrid heuristic approach to reduce the total feedback value of development projects; *Lin et al. (2015)* proposed an objective of minimizing the total feedback time based on the feedback value, and developed a local search based heuristic to optimize the project of a balancing machine; *Attari-Shendi, Saidi-Mehrabad & Gheidar-Kheljani (2019)* presented a multi-objective model that considers the total feedback value, technology risk and financial status to schedule the process of development projects. These researches have offered useful guidance in optimizing development process with interrelated activities. However, none of them considers the influence of the feedback length on the development process.

*Altus, Kroo & Gage (1995)* and *Todd (1997)* and many other studies have pointed that the feedback spanning across more activities usually leads to more reworks, which indicates

that the length of feedback may significantly affect the development progress. Therefore, a more reasonable objective of finding the activity sequence with the minimum total feedback length is proposed. Over the years, many practical applications have confirmed that the feedback length minimization (FLMP) is an appropriate approximation of minimizing the project duration, costs and risks (see, *e.g.*, *Meier, Yassine & Browning, 2007*; *Lancaster & Cheng, 2008*; *Qian & Yang, 2014*).

Due to the NP-hard nature of FLMP, it is extremely difficult to find the optimal activity sequence, even for small-scale problems. Thus, researchers turned to develop heuristic approaches to obtain near-optimal solutions. In particular, *Lin et al. (2018)* proposed an effective hybrid algorithm by integrating an insertion-based heuristic with simulated annealing. *Khanmirza, Haghbeigi & Yazdanjue (2021)* introduced the imperialist competitive algorithm to solve large-scale FLMP, which is enhanced by adaptively applying operators and tuning parameters. *Wen et al. (2021)* introduced an insertion-based heuristic algorithm (IBH) to solve a closely related problem that minimizes the total rework time. This algorithm follows the sequential improvement strategy to select operators, and experiments showed that IBH was competitive in scheduling interrelated activities. Most recently, *Peykani et al. (2023)* successively optimized the feedback length and project duration by a genetic algorithm based hybrid approach, in order to reschedule development project in resource constrained scenarios. These algorithms can obtain a reasonable solution in a short time, but cannot guarantee the global optimum.

As for exact approaches, only three studies focus on scheduling interrelated activities optimally. *Qian & Lin (2013)* reformulated FLMP as two equivalent linear programming models, then adopted the CPLEX MILP solver to optimally solve them. However, the largest FLMP that can be solved within 1 h is limited to 14 activities, and the performance of this approach is strongly affected by the density of DSM. *Gheidar-kheljani (2022)* proposed multi-objective model that minimizes the total feedback length and the cost of decreasing activity dependence. They applied CPLEX to solve small scale problems and designed a genetic algorithm for large problems. *Shang et al. (2019)* have proven that FLMP has optimal sub-structures, which allows the original problem to be divided into multiple subproblems. Based on this, they developed a hash-address based parallel branch-and-prune algorithm (HAPBP), which is the state of the art specialized exact approach in current literature. HAPBP divides FLMP into two subproblems, and concurrently schedules activities in forward and backward directions. This algorithm also employs a hash strategy to improve the efficiency of sequence comparison, by mapping activity sequences into hash values. Experiments confirm that HAPBP can solve FLMP up to 25 activities within 1 h. The shortcomings of this study are that the proposed parallel framework limits the algorithm to only use two cores of CPU, and the hash strategy is extremely space-consuming, which prevents HAPBP from fully utilizing available computing resources.

In summary, the studies on heuristic approach did not fully explore the structural properties of FLMP, and the existing heuristic algorithms are usually designed by using the classic heuristic frameworks. On the other hand, studies on specialized exact approaches for FLMP are quite limited, and there is clearly an urgent need for such dedicated exact algorithms capable of solving problem instances that cannot be solved by existing

approaches. Decomposing FLMP into subproblems to reduce the problem complexity, then solving them concurrently, is highly appealing approach to obtain the optimal activity sequence. However, the existing parallel framework and applied strategies do not take advantage of FLMP properties and available computing resources, which strongly limits the computational capability. To fulfill these research gaps, we propose in this work an novel parallel exact algorithm to solve FLMP. The main contributions are summarized as follows.

- We develop a double-decomposition based parallel branch-and-prune algorithm (DDPBP), which can employ all available computing resources to solve FLMP optimally. The proposed algorithm first divides FLMP into forward and backward scheduling subproblems, then decomposes subproblems into several scheduling tasks, and applies multiple CPU cores to prune unpromising subsequences. The resulting optimal subsequences are connected to be the global optimum.
- We propose two strategies to further enhance the DDPBP algorithm. The result-compression strategy is designed to reduce communication costs among parallel processes, by extracting and sending key information from numerous intermediate results. Furthermore, a novel hash-address strategy is developed to quickly compare and locate subsequences with lower space costs, which significantly accelerates the process of subsequence pruning.
- Computational experiments confirm the competitiveness of the DDPBP algorithm on 480 random FLMP instances, compared to the state-of-the-art exact approaches. In particular, the proposed algorithm increases the problem scale that can be solved exactly to 27 activities within 1 h, and significantly reduces the solving time for problems with less than 27 activities. In addition, further analyses shed light on the significant contributions of the result-compression and hash-address strategies to the performance of DDPBP.

## FLMP ANALYSIS

Decomposing the original problem into smaller subproblems is an effective way to solve complex problems (*Chen & Li, 2005*; *Shobaki & Jamal, 2015*; *Mitchell, Frank & Holmes, 2022*). In this section, we briefly introduce the properties of FLMP and the resulting prune criterion (*Shang et al., 2019*), which allows the algorithm to divide FLMP into two independent subproblems, and discard unpromising sequences effectively. All properties are mathematically proved in Appendix.

### Problem properties

Assume that a development project consists of activities $I = \{1, 2, \ldots, i, \ldots, n\}$, the activity sequence is $S = (s_1, s_2, \ldots, s_p, s_{p+1}, \ldots, s_n)$, and the total feedback length $fl = \sum_{h=1}^{n-1} \sum_{k=h+1}^{n} d_{s_h, s_k}(k - h)$. We set position $p(1 < p < n)$ as a split point, define that region $A_p = \{s_1, s_2, \ldots, s_p\}$ contains activities from position 1 to $p$, and region $B_p = \{s_{p+1}, s_{p+2}, \ldots, s_n\}$ contains activities after position $p$. Then we have feedback values $fv_p^a$ and $fv_p^b$ that are produced by the subsequences of regions $A_p$ and $B_p$, respectively.

**Property 1:** The total feedback length $fl = fv_p^a + fv_p^b$, and:

$$fv_p^a = \sum_{h=1}^{p-1} \sum_{k=h+1}^{p} d_{s_h s_k}(k-h) + \sum_{h=1}^{p} \sum_{k=p+1}^{n} d_{s_h s_k}(p+1-h) \tag{7}$$

$$fv_p^b = \sum_{h=p+1}^{n-1} \sum_{k=h+1}^{n} d_{s_h s_k}(k-h) + \sum_{h=1}^{p} \sum_{k=p+1}^{n} d_{s_h s_k}(k-p-1) \tag{8}$$

Property 1 shows the compositions of total feedback length $fl$, when the original sequence is split into two regions. Further, if we set the split point as position $p+1$ or $p-1$, then $fv_{p+1}^a$ and $fv_{p-1}^b$ can be derived from the following equations.

$$fv_{p+1}^a = fv_p^a + \sum_{h=1}^{p+1} \sum_{k=p+2}^{n} d_{s_h s_k} \tag{9}$$

$$fv_{p-1}^b = fv_p^b + \sum_{h=1}^{p} \sum_{k=p+1}^{n} d_{s_h s_k} \tag{10}$$

**Property 2:** Changing the subsequence of region $A_p (B_p)$ does not affect the value of $fv_p^b (fv_p^a)$.

Due to split point $p$, FLMP is divided into two subproblems, which minimize feedback values $fv_p^a$ and $fv_p^b$, and are related to regions $A_p$ and $B_p$, respectively. Property 2 indicates that although there exist feedbacks from region $B_p$ to $A_p$, the two subproblems are totally independent with each other.

## Prune criterion

We define that any two sequences are "similar", if they consist of the same activities, such as sequences $(1, 2, 3)$ and $(3, 1, 2)$; otherwise, they are "dissimilar", such as sequences $(1, 2, 3)$ and $(1, 2, 5)$. Based on the preceding properties, a prune criterion is proposed as follows:

**Prune criterion:** In region $A_p (B_p)$, for a subsequence $SA_p (SB_p)$, if its feedback value $fv_p^a (fv_p^b)$ is not the lowest among similar subsequences, then any sequence $S$ starting (ending) with $SA_p (SB_p)$ is not the global optimum, and $SA_p (SB_p)$ should be pruned.

For a certain pair of regions $A_p$ and $B_p$, assuming the optimal $SB_p^*$ of $B_p$ is found, then all high-quality sequences $S$ should end with $SB_p^*$. Therefore, the quality of $S$ depends on the quality of $SA_p$, or vice versa. In other words, the prune criterion holds. In addition, for each group of similar $SA_p$, only the one with lowest $fv_p^a$ is kept, the rest $(p! - 1)$ subsequences are pruned. The same is true for $SB_p$. We present an example to illustrate how the prune criterion works.

In Fig. 2, for a project with $I = \{1, 2, 3, 4, 5, 6, 7, 8\}$, set $p = 4$, $A_4 = \{1, 2, 4, 5\}$ and $B_4 = \{3, 6, 7, 8\}$, assume that the optimal $SB_4^*$ of $B_4$ is found. Since $fv_4^a = 6.2$ of $SA_4 = (2, 4, 1, 5)$ is higher than $fv_4^a = 5$ of $SA_4 = (1, 5, 4, 2)$, it can be concluded that

|  | $A_4 = \{1,2,4,5\}$ | $fv_4^a$ | $B_4 = \{3,6,7,8\}$ | $fv_4^b$ | $f$ |
|---|---|---|---|---|---|
| S1: | $SA_4 = (2,4,1,5)$ | 6.2 | $SB_4^* = (...)$ | ... | ... |
| S2: | $SA_4 = (1,5,4,2)$ | 5 | $SB_4^* = (...)$ | ... | ... |

|  | $A_4 = \{1,2,3,4\}$ | $fv_4^a$ | $B_4 = \{5,6,7,8\}$ | $fv_4^b$ | $f$ |
|---|---|---|---|---|---|
| S3: | $SA_4 = (2,1,3,4)$ | 5.9 | $SB_4^* = (...)$ | ... | ... |

**Figure 2** **An example of the prune criterion.**

sequence $S1$ is not the global optimum. However, the prune criterion does not work on $S1$ and $S3$, because $SA_4 = (1, 5, 4, 2)$ and $SA_4 = (2, 1, 3, 4)$ are dissimilar.

# DOUBLE-DECOMPOSITION BASED ALGORITHM

This section presents the details of the proposed Double-Decomposition based Parallel Branch-and-Prune (DDPBP) algorithm for solving FLMP, including the general concept, the main scheme, and key phases including task distribution and result combination.

## General concept

Double-decomposition means that DDPBP can decompose the whole sorting problem from two perspectives. Based on the properties of FLMP, the original problem is divided into two independent subproblems that are related to regions $A_p$ and $B_p$ respectively. By introducing the parallel framework, DDPBP can construct active sequences in forward (from head to tail, $A_p$) and backward (from tail to head, $B_p$) directions concurrently. Figure 3 shows the search trees applied in DDPBP. In the forward tree, each node represents a subsequence from position 1 to $p$ within a complete sequence, for example, node $(7, 6, 5)$ is the first three activities of one complete sequence, and child node $(7, 6, 5, 4)$ is built by adding activity 4 to the end of node $(7, 6, 5)$. The backward tree follows the same structure, but represents the opposite direction. DDPBP traverses two trees in a breadth-first way, along with pruning unpromising nodes (marked by red line). When the exploration finishes, the remaining partial sequences (leaf nodes) are connected as complete sequences (marked by blue line), from which we can find the optimal solution.

These two exploring processes are totally independent without any information exchange, which can be distributed to two cores of CPU. However, this framework limits the full use of available computing resources. As multi-core computers are common nowadays, a more flexible framework that supports any number of cores, is necessary. Figure 4 presents a further decomposition in the forward and backward processes. For a FLMP with seven activities, assume that six cores are available, then we can assign half of cores to each process. For the forward process, in row 3, nodes are divided into three groups and sent to three cores for node pruning. Since each core only handles partial nodes, after all tasks are finished, DDPBP gathers the results and proceeds further node pruning. After

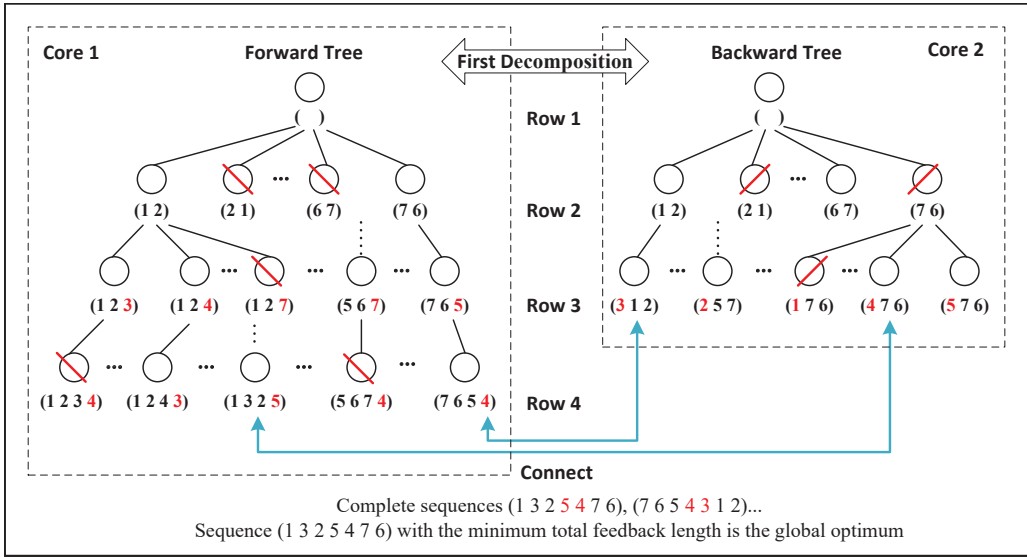

**Figure 3    Forward and backward trees for a FLMP with seven activities.**

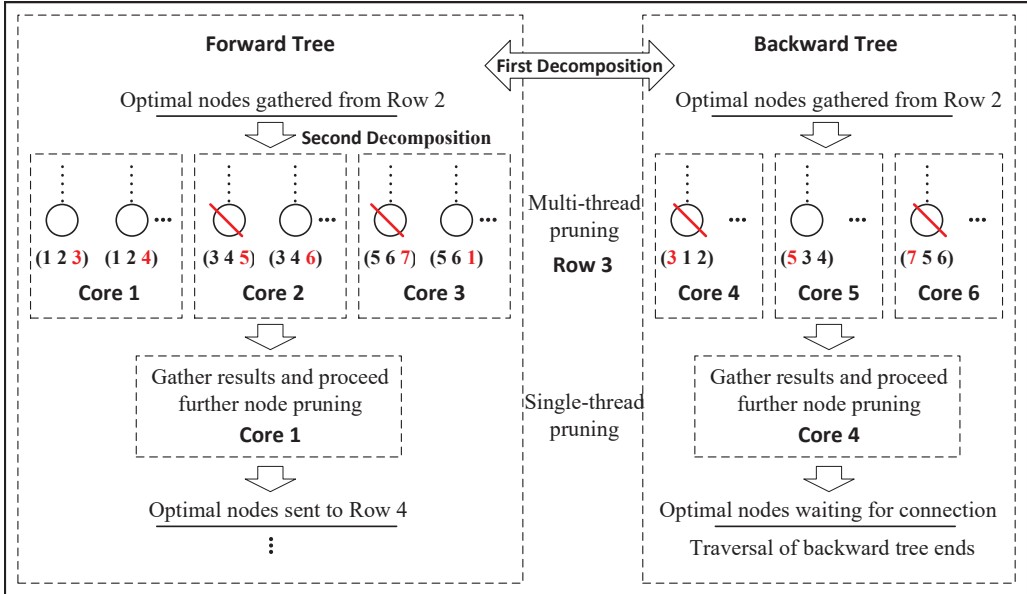

**Figure 4    Double-decomposition for a FLMP with seven activities.**

all unpromising nodes are discarded, the remaining nodes are used to generate child nodes for the next row. The decomposition in the backward process follows the same way.

The second decomposition makes it possible to take full advantage of multiple cores to share the workloads. Although forward and backward processes do not communicate with each other, multiple threads within two processes still exchange data frequently. Researches show that the communication cost in parallel frameworks cannot be ignored

(*Tsai et al., 2021*; *Wang & Joshi, 2021*). Therefore, how to reduce the impact of multi-thread communication is an important issue in this study.

---

**Algorithm 1: Main scheme.**

**Input:** $D = (d_{ij})_{n \times n}$, $cn$, $na$;

/* Design structure matrix, core number, workloads of forward process */

**Output:** $fl^*$, $S^*$;

**Task distribution phase:**

| | | | | | |
|---|---|---|---|---|---|
| /* Traverse the forward tree */ | | | /* Traverse the backward tree */ | | |
| **1** | **For** row $p = 2 : 1 : na$ | | **1** | **For** row $p = (n-3) : -1 : na$ | |
| | 1.1 | **If** (Backward process finishes) | | 1.1 | **If** (Forward process finishes) |
| | | $cna = cn$; | | | $cnb = cn$; |
| | | **Else** | | | **Else** |
| | | $cna = cn/2$; | | | $cnb = cn/2$; |
| | 1.2 | $SetA_p \leftarrow$ Forward-process | | 1.2 | $SetB_p \leftarrow$ Backward-process |
| | | $(D, SetA_{p-1}, cna)$; | | | $(D, SetB_{p+1}, cnb)$; |
| **1** | **End for** | | **1** | **End for** |

**Result combination phase:**

/* Switches to the single thread */

**2**    Connect all $SA_{na}$ in $SetA_{na}$ with corresponding $SB_{na}$ in $SetB_{na}$, and set $fl = fv_{na}^a + fv_{na}^b$;

**3**    Return the optimal sequence $S^*$ with the lowest feedback length $F^*$.

---

## Main scheme

Algorithm 1 presents the main scheme of DDPBP. The whole procedure consists of a task distribution phase ('Task distribution phase') and a result combination phase ('Result combination phase'). For a FLMP problem with $n$ activities, assume that there are $cn$ cores available. Starting with a given parameter $na$, the algorithm sets the number of rows that the forward process needs to explore as $na$, and sets the number of rows explored by the backward process as $(n - na)$. Then, the task distribution phase concurrently traverses the forward and backward trees row by row, and applies the forward and backward process to discard unpromising nodes (Step 1). Since $na$ and $(n - na)$ may be not equal, if both processes are running, the algorithm distributes cores equally to two processes ($cn/2$ for each); if one process ends earlier, the remaining process adaptively takes all the cores to make a full use of computing resources (Step 1.1). After tree explorations finish, in the result combination phase, each partial sequence $SA_{na}$ contained in $SetA_{na}$ is connected to its corresponding sequence $SB_{na}$ in $SetB_{na}$ to construct the complete sequence. Finally, the one with the minimum feedback length among all complete sequences is the global optimum (Step 2–3).

## Task distribution phase

The task distribution phase realizes the double decomposition of the FLMP problem. The first decomposition is to concurrently schedule activities in forward and backward

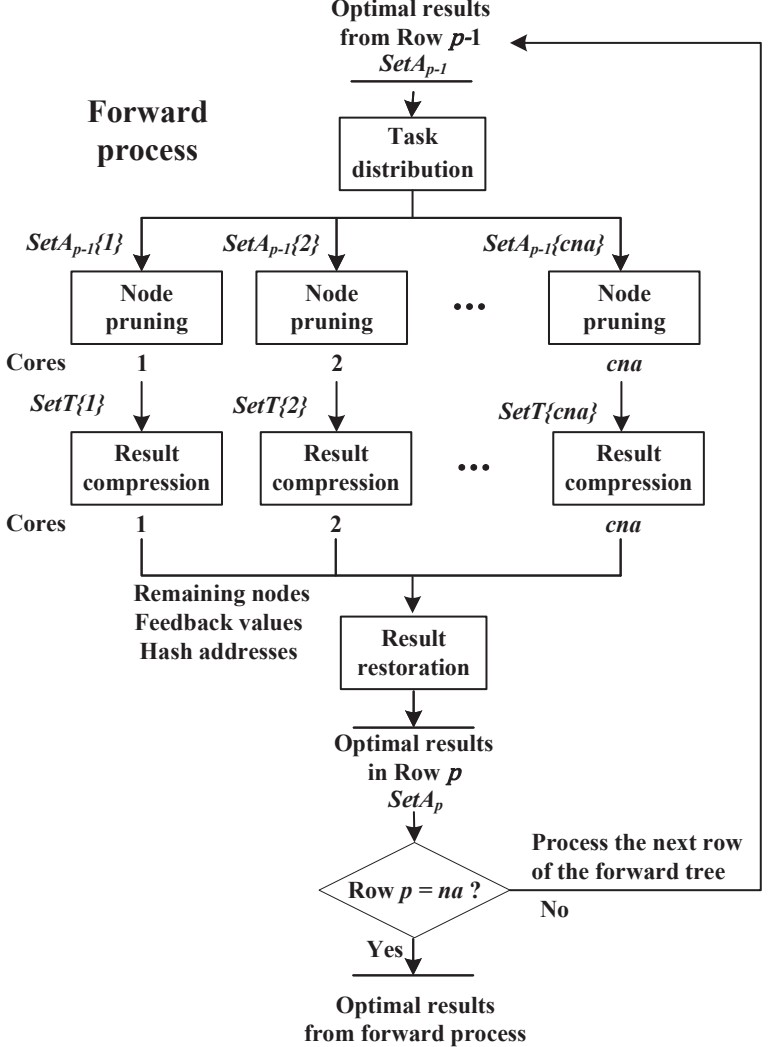

**Figure 5  Forward process.**

directions. The second decomposition is to distribute pruning tasks of each row to given cores within forward and backward processes. We use the forward process as an example to illustrate this idea, and the backward process follows the same procedure except exploring the backward tree.

As shown in Fig. 5, the forward process consists of four components, including task distribution, node pruning, result compression and result restoration. These components work sequentially on each row until reaching row $p = na$.

**Task distribution:** Suppose that *cna* cores are available. The algorithm receives $n_{p-1}$ nodes stored in $SetA_{p-1}$ from row $p - 1$, and is about to explore row $p$. Then these nodes are divided into *cna* equal parts, *i.e.*, $SetA_{p-1}\{i\}(1 \leq i \leq cna)$ with $n_{p-1}/cna$ nodes, and sends them to *cna* cores respectively. This procedure is single thread.

**Node pruning:** As shown in Algorithm 2, in the forward process, assume that core $i$ is exploring row $p$ and receives partial nodes stored in $SetA_{p-1}\{i\}$. Step 1 adds a new activity to the end of node $SA_{p-1}$ to build child node $SA_p$, and calculates $fv_p^a$ using recursive Eq. (9) (if row $p = 2$, use Eq. (7) instead); Steps 3.1–3.2 locate the similar node of $SA_p$ at $SetT\{i\}(ha)$ and only save the node with lower $fv_p^a$ at this position, where $SetT\{i\}$ is a temporary result set for core $i$ and $ha$ is a unique hash address for each group of similar nodes (see in 'Hash strategy'). These steps repeat, until all the child nodes $SA_p$ are checked.

**Result compression:** The hash-address strategy is introduced to boost the efficiency of searching similar nodes in $SetT\{i\}$. In 'Hash strategy', we propose hash functions to map each group of similar nodes into a unique hash address $ha$. In order to support all possible addresses, the size of $SetT\{i\}$ is set as $C_n^p$, which equals to the total number of similar node groups in row $p$. However, since a core only handles a partial task, it does not need to use all the space of $SetT\{i\}$. In fact, the hash addresses appearing in a core are usually discrete and irregular, such as $\{1, 3, \ldots, 20, 26\}$, which causes the final $SetT\{i\}$ to be sparse. Hence, in order to reduce communication cost, after node pruning finishes, the algorithm extracts remaining nodes, corresponding $fv_p^a$ and hash addresses from $SetT\{i\}$, and sends them to the next component, instead of transmitting the entire $SetT\{i\}$ (detailed analysis in 'Effectiveness of result-compression strategy').

---

**Algorithm 2: Node pruning.**

**Input:** $D = (d_{ij})_{n \times n}$, $SetA_{p-1}\{i\}$;

/* Design structure matrix, Partial results from row $p-1$ */

**Output:** $SetT\{i\}$;

| | | |
|---|---|---|
| 1 | Based on $SetA_{p-1}\{i\}$, construct child nodes $SA_p$ for parent nodes $SA_{p-1}$; | |
| 2 | **If ( $p = 2$)** | |
| | Calculate $fv_p^a$ for each $SA_p$ using Eq. 7; | |
| | **Else** | |
| | Calculate $fv_p^a$ for each $SA_p$ using Eq. 9; | |
| 3 | **While** (There exists an unchecked $SA_p$) | |
| | **3.1** | Select an unchecked $SA_p$, and calculate hash address $ha$ using Eq. 11; |
| | **3.2** | **If** $SetT\{i\}(ha)$ is empty |
| | | Save $SA_p$ and $fv_p^a$ at $SetT\{i\}(ha)$; |
| | | **Else** |
| | | Compare $fv_p^a$, and save the one with lower $fv_p^a$ at $SetT\{i\}(ha)$; |
| 3 | **End while** | |
| 4 | **Return** $SetT\{i\}$ | |

---

**Result restoration:** After receiving nodes, $fv_p^a$ and hash addresses from multiple cores, the whole process switches from multi threads to single thread. For the results from core $i$, according to hash addresses $ha$, the algorithm assigns nodes and $fv_p^a$ to $SetA_p(ha)$, where $SetA_p$ with the size of $C_n^p$, contains the optimal node among each group of similar nodes in row $p$. If $SetA_p(ha)$ is not empty, the algorithm keeps the one with lower $fv_p^a$ in this

position, to further prune unpromising nodes. This procedure repeats until all results from different cores are checked. Then, the algorithm is ready to explore row $p+1$.

### Result combination phase

After the task distribution phase finishes, the algorithm connects the nodes $SA_{na}$ in $SetA_{na}$ with the corresponding nodes $SB_{na}$ in $SetB_{na}$ to construct the complete activity sequences, and calculate the total feedback length by $fl = fv_{na}^a + fv_{na}^b$, from which the sequence with minimum total feedback length is the global optimum.

In addition, since identifying and searching the right $SB_{na}$ in $SetB_{na}$ for each $SA_{na}$ are time-consuming, we introduce a hash strategy to improve the efficiency of the combination phase. In 'Hash strategy', we propose a function Eq. (15) that can derive the hash address of $SB_{na}$ from the hash address of $SA_{na}$. Hence, when the algorithm receives a node $SA_{na}$ with hash address $ha_a$ from $SetA_{na}$, the corresponding node $SB_{na}$ can be easily located at position $SetB_{na}(ha_b)$.

### Computational complexity

We first consider the task distribution phase. For a FLMP with $n$ activities, we assign $na$ and $n - na$ rows to forward and backward processes, respectively. Due to the parallel nature of DDPBP, the computational complexity of the process that explores more rows can represent the whole algorithm. Without loss of generality, we set $na > n - na$ and take the forward process as an example. For any row $p(1 < p \leq na)$, the number of nodes needed to be processed is $C_n^{p-1} * (n-p+1)$. Hence, the complexity of exploring the forward tree is $O(\sum_{p=2}^{na} C_n^{p-1} * (n-p+1))$, where $C_n^p = n!/(p! * (n-p)!)$.

We now consider the result combination phase. At the beginning, there are $C_n^{na}$ pairs of nodes needed to be connected. With the help of hash address, DDPBP can locate and connect right nodes directly. Hence the complexity of this phase is $O(C_n^{na})$.

In addition, due to the similar structure of search trees, the overall complexity of DDPBP is quite close to the complexity of HAPBP (*Shang et al., 2019*), which is $O(\sum_{p=2}^{\lfloor n/2 \rfloor} C_n^{p-1} * (n-p+1))$. However, double-decomposition framework allows the proposed algorithm to applied more computational resources in the search process.

## HASH STRATEGY

During the forward and backward processes, the algorithm needs to find the similar nodes in $SetT$ for each node of row $p$; however, the time complexity of determining whether two nodes are similar is $O(n^2)$, and locating the similar nodes in $SetT$ is $O(|SetT| * n^2)$, in the worst case. Hence, it is necessary to convert nodes into hash values, and perform hash value comparison instead of node comparison to boost the efficiency.

*Shang et al. (2019)* applied the hash-address strategy in the HAPBP algorithm, which transforms each group of similar nodes into a unique hash address in a result $Set$, by using hash function $ha = \sum_{i \in SA_p(SB_p)} 2^{i-1}$. HAPBP can find the similar nodes for any node at $Set(ha)$ directly. However, as shown in Fig. 6, this function allocates hash address for all groups of similar nodes in the search tree, no matter which rows these nodes belong to. In fact, the space complexity of this strategy is $O(\sum_{p=2}^{\lfloor n/2 \rfloor} C_n^p)$, which causes $Set$ to

| ha | $SA_p(1 < p < n)$ |
|----|-------------------|
| 3 | $SA_2 = (1,2)$ |
| 23 | $SA_4 = (2,5,1,3)$ |
| 23 | $SA_4 = (1,2,5,3)$ |
| 39 | $SA_4 = (1,2,6,3)$ |
| 55 | $SA_5 = (1,5,2,6,3)$ |

**Figure 6  HAPBP's hash strategy.**

support all addresses along with corresponding storage spaces, and become extremely space-consuming. It is difficult to maintain and transmit such a huge array in a parallel framework.

This study presents a new hash strategy that only maps similar nodes within a row to a consecutive hash address $ha$ $(1 \le ha \le C_n^p)$, which has a space complexity of $O(C_n^{\lfloor n/2 \rfloor})$ in the worst case, and can significantly reduce space costs of $SetT$. For any node in row $p$:

$$SA_p = (s_1, s_2, \ldots, s_i, \ldots, s_j, \ldots, s_p)$$

We first reorder $SA_p$ as $s_i < s_j (1 \le i < j \le p)$, then encode it as a hash address by the following hash functions:

$$ha = h(s_1) + \sum_{i=2}^{p-1} h(s_i) + h(s_p) \tag{11}$$

$$h(s_1) = \begin{cases} \sum_{t=1}^{s_1 - 1} C_{n-t}^{p-1}, & s_1 \neq 1 \\ 0, & s_1 = 1 \end{cases} \tag{12}$$

| ha | $SA_3$ | ha | $SA_3$ | Similar $SA_3$ |
|----|--------|----|--------|----------------|
| 1 | (1,2,3) | 6 | (1,4,5) | (2,4,5) |
| 2 | (1,2,4) | 7 | (2,3,4) | (2,5,4) |
| 3 | (1,2,5) | 8 | (2,3,5) | ... ... |
| 4 | (1,3,4) | 9 | (2,4,5) ← sort | (5,2,4) |
| 5 | (1,3,5) | 10 | (3,4,5) | (5,4,2) |

**Figure 7 Numerical illustrations for the hash strategy.**

$$h(s_i) = \begin{cases} \sum\limits_{t=s_{i-1}+1}^{s_i-1} C_{n-t}^{p-i}, & s_{i-1}+1 \neq s_i \\ 0, & s_{i-1}+1 = s_i \end{cases} \tag{13}$$

$$h(s_p) = s_p - s_{p-1} \tag{14}$$

where $ha$ is unique for each group of similar nodes in row $p$. Therefore, when reaching any node $SA_p$ in row $p$, we can find its similar nodes at $SetT(ha)$ and compare them directly. We present an example $\{1,2,3,4,5\}$ with $n=5, p=3$ to illustrate this idea.

As shown in Fig. 7, each group of similar nodes in row 3 is mapped to a unique $ha(1 \leq ha \leq 10)$, and $SetT$ only contains the information of row 3, which is quite space-saving and easy to split among different cores. Similar nodes (marked in red) are first reordered as $(2,4,5)$, then Eq. (11) converts $SA_3 = (s_1, s_2, s_3) = (2,4,5)$ into $ha=9$ as follows:

$$ha_{(2,4,5)} = h(s_1) + h(s_2) + h(s_3) = \sum_{t=1}^{2-1} C_{5-t}^{3-1} + \sum_{t=2+1}^{4-1} C_{5-t}^{3-2} + 1 = C_4^2 + C_2^1 + 1 = 9$$

For $SA_3 = (1, 4, 5)$, $SA_3 = (1, 2, 4)$, we can achieve their *ha* as follows:

$$ha_{(1,4,5)} = h(s_1) + h(s_2) + h(s_3) = 0 + \sum_{t=2}^{3} C_{5-t}^1 + 1 = 0 + C_3^1 + C_2^1 + 1 = 6$$

$$ha_{(1,2,4)} = h(s_1) + h(s_2) + h(s_3) = 0 + 0 + (4 - 2) = 2$$

Since Eq. (11) is performed frequently during the search process, we can calculate combination number $C_n^m$ before the search process starts. The algorithm just selects appropriate values from a predefined array according to $(m, n)$, instead of recalculating $C_n^m$.

The hash strategy is also applied to accelerate the combination phase. For any node $SA_{na}$ in $SetA_{na}$ with a hash address $ha_a$, the algorithm can use the following function to derive the hash address $ha_b$ of the corresponding node $SB_{na}$ in $SetB_{na}$:

$$ha_b = C_n^{na} + 1 - ha_a \tag{15}$$

For instance, suppose that $n = 6$, $na = 3$, $SA_3 = (4, 2, 1)$ and $SB_3 = (5, 3, 6)$. After resorted two nodes, we can apply Eq. (11) to obtain $ha_{(4,2,1)} = 2$ and $ha_{(5,3,6)} = 19$. Based on Eq. (15), we achieve that $ha_{(5,3,6)} = C_6^3 + 1 - ha_{(4,2,1)} = 21 - 2 = 19$. In other words, Eq. (15) holds.

# COMPUTATIONAL EXPERIMENTS

This section reports computational experiments to evaluate the effectiveness of the DDPBP algorithm. Specifically, we first describe the benchmark instances and the experimental protocol. Then, we make comparisons between the proposed algorithm and state of the art algorithms in literature.

## Benchmark instances and experimental protocol

We use random DSM with various sizes and densities as benchmark instances. For each DSM, the degree of information dependence $(d_{i,j})$ follows uniform distribution, and the density level is the ratio of non zero elements. A DSM generator is introduced to produce random instances (*Qian & Lin, 2013*), where the number of activities ($n$) is set as $\{15, 17, 19, 21, 23, 25, 26, 27\}$, and the density of DSM (*den*) is set as $\{0.1, 0.2, 0.4, 0.6, 0.8, 1\}$. For each pair of $n$ and *den*, 10 instances are generated, leading to a total number of 480 instances used in the experiments.

The DDPBP algorithm is coded in MATLAB 2018 (MathWorks, Natick, MA, USA) with the Parallel Computing Toolbox and runs under the recommended setting of $\{cn = 8, na = 5\}$ ('Parameter analysis'). The algorithms for comparisons include: the HAPBP algorithm from *Shang et al. (2019)*, the branch and cut algorithm and the branch and bound algorithm of the CPLEX and Gurobi solvers. All experiments are conducted on a Lenovo laptop with a 2.90 GHz AMD Ryzen 7 processor (8 cores) and a 64 GB RAM.

Two kinds of experiments are conducted. The first one is the comparison between DDPBP and state of the art exact algorithms ('Comparisons of DDPBP with exact algorithms'). We report the average times of obtaining the optimal solutions, and the

gap information of objective values. The two-tailed sign test is introduced to determine if there exists statistical differences between the performances of two algorithms (*Demšar, 2006*; *Shang et al., 2023*). For each pairwise comparison with $N$ tests, if one algorithm wins at least $CV^N_{0.05} = N/2 + 1.96\sqrt{N}/2$ times, then this algorithm performs significantly better than the other one at the level of 0.05.

The second experiment compares DDPBP with two heuristic algorithms ('Comparisons of DDPBP with heuristic algorithms'), which aims to know whether these heuristic algorithms can reach the global optimum compared to the optimal objective values obtained by our DDPBP algorithm.

## Comparisons of DDPBP with exact algorithms

This section presents detailed comparisons between the proposed DDPBP algorithm and three representative exact algorithms, including: the HAPBP algorithm that is the best dedicated algorithm for FLMP in the literature; the branch and cut based CPLEX 12.6 MILP solver and the branch and bound based Gurobi 10.1 MILP solver, which have been widely applied in solving NP-hard problems, and have an exponential complexity. In this experiment, the two general solvers are set to parallel mode to allow them to use all available cores of the computer. For each pair of activity number $n$ and density level *den*, four compared algorithms solve 10 random instances. The average solving times and objective value gaps are reported in Tables 2 and 3, respectively.

Table 2 presents the average solving times over these 10 instances for each FLMP setting, where the mark "−" means that the algorithm can not obtain the optimal solutions for this kind of instances within 1 h. The results indicate that DDPBP outperforms the other three algorithms. Compared to HAPBP, the proposed algorithm spends less times for all types of instances, and increases the scale of FLMP that the dedicated exact algorithm can solve within 1 h from 25 activities to 27 activities. It confirms the effectiveness of double-decomposition strategy, result-compression strategy and hash-address strategy. In terms of general solvers, DDPBP performs significantly better than CPLEX and Gurobi MILP solvers with 42 and 47 better results respectively, according to two-tailed sign test $(42, 47 > CV^{48}_{0.05} \approx 31)$. We observe that the general solvers usually perform better when the density levels of DSM are very low. However, as the density level increases, the time consumptions for problem solving increase rapidly. For example, for instances with 17 activities and $\{0.2, 0.4, 0.6\}$ density levels, the average solving times of Gurobi are 23.43 s, 219.74 s and 2406.03 s, respectively. Meanwhile, the solving times of DDPBP are 0.62 s, 0.79 s and 0.55 s. This experiment demonstrates that the prune criterion are not affected by density level, which makes DDPBP more stable and applicable for FLMP.

Table 3 shows the average gaps of objective values over 10 instances for each FLMP setting. In this table, "*opt*" represents the average optimal objective value obtained by DDPBP, "*o_gap*" describes the average gap between the optimal objective value (*opt*) and the objective value (*obj*) obtained by other algorithms ($(obj\_opt)/opt*100$). "*b_gap*" represents the average gap between the objective value (*obj*) and the best bound (*bou*), which is reported by the two general solvers ($(obj - bou)/obj*100$) and indicates the convergence status when solvers stop. In addition, since DDPBP and HAPBP apply the

**Table 2  Comparison of average solving times (seconds).**

| n | den | DDPBP | HAPBP | CPLEX | Gurobi | n | den | DDPBP | HAPBP | CPLEX | Gurobi |
|---|---|---|---|---|---|---|---|---|---|---|---|
| | 0.1 | 0.35 | 0.74 | **0.17** | 0.33 | | 0.1 | 0.71 | 2.85 | **0.52** | 1.14 |
| | 0.2 | **0.46** | 0.7 | 0.51 | 6.79 | | 0.2 | **0.62** | 2.87 | 5.47 | 23.43 |
| 15 | 0.4 | **0.42** | 0.7 | 11.03 | 31.08 | 17 | 0.4 | **0.79** | 2.86 | 243.57 | 219.74 |
| | 0.6 | **0.33** | 0.7 | 43.61 | 72.7 | | 0.6 | **0.55** | 2.86 | 891.85 | 2406.03 |
| | 0.8 | **0.45** | 0.69 | 264.59 | 498.71 | | 0.8 | **0.61** | 2.85 | – | – |
| | 1 | **0.41** | 0.69 | – | 1620.88 | | 1 | **0.8** | 2.85 | – | – |
| | 0.1 | 1.83 | 12.62 | **1.06** | 3.51 | | 0.1 | 7.2 | 63.47 | **3.34** | 23.35 |
| | 0.2 | **1.78** | 12.53 | 32.43 | 100.44 | | 0.2 | **7.24** | 61.32 | 434.69 | 879.57 |
| 19 | 0.4 | **1.83** | 12.56 | – | 1711.97 | 21 | 0.4 | **7.26** | 60.14 | – | – |
| | 0.6 | **1.95** | 13.84 | – | – | | 0.6 | **7.68** | 58.47 | – | – |
| | 0.8 | **1.8** | 13.54 | – | – | | 0.8 | **8.34** | 61.29 | – | – |
| | 1 | **1.82** | 14.07 | – | – | | 1 | **9.25** | 61.49 | – | – |
| | 0.1 | 38.23 | 269.09 | **22.28** | 56.88 | | 0.1 | **170.3** | 1142.31 | 378.89 | 213.19 |
| | 0.2 | **33.05** | 270.1 | – | 1960.3 | | 0.2 | **170.62** | 1128.52 | – | – |
| 23 | 0.4 | **32.96** | 263.24 | – | – | 25 | 0.4 | **164.91** | 1165.72 | – | – |
| | 0.6 | **38.28** | 267.82 | – | – | | 0.6 | **165.05** | 1181.56 | – | – |
| | 0.8 | **38.29** | 262.25 | – | – | | 0.8 | **143.71** | 1148.83 | – | – |
| | 1 | **38.27** | 264.77 | – | – | | 1 | **142.47** | 1150.88 | – | – |
| | 0.1 | **325.47** | 2135.33 | 492.75 | 1146.93 | | 0.1 | 774.7 | – | **737.47** | 1856.91 |
| | 0.2 | **303.63** | 2109.73 | – | – | | 0.2 | **716.97** | – | – | – |
| 26 | 0.4 | **297.27** | 2350.52 | – | – | 27 | 0.4 | **785.23** | – | – | – |
| | 0.6 | **298.39** | 2344.28 | – | – | | 0.6 | **682.64** | – | – | – |
| | 0.8 | **296.15** | 2108.32 | – | – | | 0.8 | **713.73** | – | – | – |
| | 1 | **338.44** | 2290.08 | – | – | | 1 | **683.88** | – | – | – |

**Notes.**

Best results are shown in bold.

breadth first strategy to traverse search trees, they cannot achieve feasible solutions until the search finishes. Thus, in Table 3, DDPBP and HAPBP do not have column " $b\_gap$" and the resulting solution is the global optimum.

As shown in Table 3, HAPBP cannot achieve the optimal solution of FLMP with 27 activities in 1 h (marked by "–"). As for the general solvers, CPLEX can obtain the optimal solutions for 17 out of 48 kinds of instances ($o\_gap = 0$), most of which have a low activity number and low density. For example, for FLMP with 27 activities and 0.1 density level, CPLEX achieves the global optimum within 1 h. However, when the density level increases to 0.2, the average gap between a feasible solution and the global optimum is $o\_gap = 13.02\%$, and the average bound gap is $b\_gap = 58.40\%$, which is quite large. On the other hand, the quality of feasible solutions obtained by Gurobi are much better. In fact, some of them are actually the global optimum ($o\_gap = 0, b\_gap \neq 0$, 10 kinds of instances), compared to the optimal results from DDPBP. However, it is difficult for Gurobi to prove the global optimum within 1 h, since the corresponding bound gaps $b\_gap$ are still very high.

Shang et al. (2023), *PeerJ Comput. Sci.*, DOI 10.7717/peerj-cs.1597

**Table 3  Comparison of average objective values.**

| n | den | DDPBP | HAPBP | CPLEX | | Gurobi | | n | den | DDPBP | HAPBP | CPLEX | | Gurobi | |
|---|---|---|---|---|---|---|---|---|---|---|---|---|---|---|---|
| | | opt | o_gap | o_gap | b_gap | o_gap | b_gap | | | opt | o_gap | o_gap | b_gap | o_gap | b_gap |
| | 0.1 | 1.40 | 0 | 0 | 0 | 0 | 0 | | 0.1 | 3.00 | 0 | 0 | 0 | 0 | 0 |
| | 0.2 | 12.75 | 0 | 0 | 0 | 0 | 0 | | 0.2 | 21.45 | 0 | 0 | 0 | 0 | 0 |
| 15 | 0.4 | 50.99 | 0 | 0 | 0 | 0 | 0 | 17 | 0.4 | 77.23 | 0 | 0 | 0 | 0 | 0 |
| | 0.6 | 99.37 | 0 | 0 | 0 | 0 | 0 | | 0.6 | 145.63 | 0 | 0 | 0 | 0 | 0 |
| | 0.8 | 147.61 | 0 | 0 | 0 | 0 | 0 | | 0.8 | 232.24 | 0 | 0.34 | **25.45** | **0** | 38.58 |
| | 1 | 210.82 | 0 | 3.07 | 22.43 | **0** | **0** | | 1 | 317.70 | 0 | 7.68 | **46.10** | **0** | 47.83 |
| | 0.1 | 3.41 | 0 | 0 | 0 | 0 | 0 | | 0.1 | 6.30 | 0 | 0 | 0 | 0 | 0 |
| | 0.2 | 32.86 | 0 | 0 | 0 | 0 | 0 | | 0.2 | 42.47 | 0 | 0 | 0 | 0 | 0 |
| 19 | 0.4 | 117.96 | 0 | 2.60 | 35.37 | **0** | **0** | 21 | 0.4 | 151.13 | 0 | 3.22 | **35.33** | **0** | 56.56 |
| | 0.6 | 214.97 | 0 | 5.53 | **43.13** | **0** | 50.63 | | 0.6 | 290.55 | 0 | 7.46 | **45.63** | **0** | 70.96 |
| | 0.8 | 321.96 | 0 | 13.49 | **43.03** | **0** | 64.85 | | 0.8 | 450.37 | 0 | 9.07 | **55.25** | 8.03 | 76.21 |
| | 1 | 450.20 | 0 | 2.32 | **39.79** | 0.36 | 65.52 | | 1 | 615.38 | 0 | 12.56 | **59.77** | **0** | 76.20 |
| | 0.1 | 10.59 | 0 | 0 | 0 | 0 | 0 | | 0.1 | 12.88 | 0 | 0 | 0 | 0 | 0 |
| | 0.2 | 58.11 | 0 | 9.03 | 28.45 | **0** | **0** | | 0.2 | 80.89 | 0 | 37.81 | **49.02** | 8.83 | 65.86 |
| 23 | 0.4 | 204.75 | 0 | 5.13 | **50.25** | 1.21 | 70.61 | 25 | 0.4 | 290.13 | 0 | **1.75** | **68.20** | 3.23 | 74.35 |
| | 0.6 | 407.99 | 0 | 2.00 | **61.89** | **0** | 86.51 | | 0.6 | 522.72 | 0 | **0.05** | **62.12** | 2.88 | 79.74 |
| | 0.8 | 577.35 | 0 | 3.92 | **66.72** | **0** | 79.52 | | 0.8 | 773.15 | 0 | 1.97 | **70.43** | 1.72 | 80.28 |
| | 1 | 817.92 | 0 | 4.82 | **76.28** | **0** | 80.56 | | 1 | 1074.76 | 0 | **6.46** | **80.41** | 9.93 | 90.48 |
| | 0.1 | 18.68 | 0 | 0 | 0 | 0 | 0 | | 0.1 | 25.52 | – | 0 | 0 | 0 | 0 |
| | 0.2 | 101.98 | 0 | 1.83 | **45.92** | 1.27 | 57.90 | | 0.2 | 121.95 | – | 13.02 | **58.40** | 12.71 | 65.56 |
| 26 | 0.4 | 323.20 | 0 | 19.32 | **68.24** | 15.34 | 76.95 | 27 | 0.4 | 365.00 | – | 16.11 | **79.07** | 4.53 | 82.03 |
| | 0.6 | 594.95 | 0 | **0.41** | **61.88** | 0.79 | 76.82 | | 0.6 | 665.93 | – | 13.76 | **79.94** | 2.52 | 83.60 |
| | 0.8 | 920.61 | 0 | 4.28 | **71.67** | 2.31 | 86.49 | | 0.8 | 1011.61 | – | 3.46 | **80.40** | 1.18 | 86.82 |
| | 1 | 1195.72 | 0 | 5.68 | **76.53** | 3.57 | 89.11 | | 1 | 1327.83 | – | **1.25** | **72.58** | 11.98 | 90.71 |

**Notes.**
Best results are shown in bold.

## Comparisons of DDPBP with heuristic algorithms

Since DDPBP can provide the optimal solutions of FLMP with up to 27 activities, it is worthwhile to use DDPBP as a benchmark to evaluate the performance of heuristic approaches, especially to see if heuristic algorithms can obtain the global optimum. In this section, we introduce two state-of-the art algorithms to solve the instances from Section 'Benchmark instances and experimental protocol'. The first one is the insertion-based heuristic algorithm(IBH) (*Wen et al., 2021*), which follows the local search framework and apply multiple operators including activity insertion and activity block insertion. The second algorithm is the multi-wave tabu search (MWTS) algorithm (*Shang et al., 2023*), which alternates between a tabu-search based intensification phase and a hybrid perturbation phase. The computational complexity of both algorithms is $O(n^2)$, which is much lower than $O(\sum_{p=2}^{na} C_n^{p-1} * (n-p+1))$ of DDPBP. We implemented these algorithms on the Matlab platform, and set the time limits as 6 min.

Table 4 reports the average gaps of the objective values obtained by these heuristic algorithms and the optimal values from DDPBP. We observe that IBH actually reaches the global optimum for 15 out of 48 kinds of instances ($o\_gap = 0$), compared to the existing optimal objective values (*opt*). However, as the number of activities in FLMP increases, it is more difficult for IBH to achieve the optimal solutions. For example, for FLMP with 25 activities and 0.2 density level, the average gap between feasible solutions and the global optimum is $o\_gap = 3.91\%$. For MWTS, it performs significantly better for obtaining the optimal solutions of all instances, which confirms the strong intensification ability of tabu search, and the necessity of applying a perturbation strategy for diversification in solving FLMP. This experiment inspires us to apply tabu-search in the parallel exact algorithm to efficiently generate good bound and cut search branches. On the other hand, decomposing a complex problem into subproblems, then apply tabu search to solve them concurrently, may lead to an effective heuristic framework for solving complex problems with large scale.

## ANALYSIS

This section provides systematic analyses for parameters and strategies applied in the algorithm. We first conduct a sensitivity analysis to see if there exist significant differences among different parameter settings. Then, to confirm the effectiveness of double-decomposition strategy, result-compression strategy and hash-address strategy, DDPBP is compared with three variants whose related components are removed.

### Parameter analysis

The proposed algorithm is controlled by parameters $cn$ and $na$. Parameter $cn$ represents the number of cores that are utilized by the algorithm, and the default value is 8 which is the maximum number of available cores in our computer. Parameter $na$ is the number of rows explored by the forward process, and the recommended setting is 5, which is used to adjust workloads of two processes.

The instances are set as $\{18, 20, 22, 24\}$ activities and $\{0.1, 0.5, 0.9\}$ density level. The ranges of $cn$ and $na$ are $\{2, 4, 6, 8\}$ and $\{3, 5, 7, 9, 11, 13, 15\}$ respectively. For each parameter,

**Table 4  Comparison of DDPBP and heuristic algorithms.**

| n | den | DDPBP | IBH | MWTS | n | den | DDPBP | IBH | MWTS |
|---|-----|-------|-----|------|---|-----|-------|-----|------|
|   |     | opt | o_gap | o_gap |   |     | opt | o_gap | o_gap |
| 15 | 0.1 | 1.40 | 0 | 0 | 17 | 0.1 | 3.00 | 0 | 0 |
|    | 0.2 | 12.75 | 0 | 0 |    | 0.2 | 21.45 | 6.20 | **0** |
|    | 0.4 | 50.99 | 0 | 0 |    | 0.4 | 77.23 | 2.59 | **0** |
|    | 0.6 | 99.37 | 0 | 0 |    | 0.6 | 145.63 | 0 | 0 |
|    | 0.8 | 147.61 | 0 | 0 |    | 0.8 | 232.24 | 0 | 0 |
|    | 1 | 210.82 | 0 | 0 |    | 1 | 317.70 | 0.32 | **0** |
| 19 | 0.1 | 3.41 | 34.83 | **0** | 21 | 0.1 | 6.30 | 16.62 | **0** |
|    | 0.2 | 32.86 | 0 | 0 |    | 0.2 | 42.47 | 0 | 0 |
|    | 0.4 | 117.96 | 0 | 0 |    | 0.4 | 151.13 | 0.80 | **0** |
|    | 0.6 | 214.97 | 0.41 | **0** |    | 0.6 | 290.55 | 1.40 | **0** |
|    | 0.8 | 321.96 | 0.62 | **0** |    | 0.8 | 450.37 | 0.14 | **0** |
|    | 1 | 450.20 | 0.25 | **0** |    | 1 | 615.38 | 0.28 | **0** |
| 23 | 0.1 | 10.59 | 12.57 | **0** | 25 | 0.1 | 12.88 | 11.45 | **0** |
|    | 0.2 | 58.11 | 2.06 | **0** |    | 0.2 | 80.89 | 3.91 | **0** |
|    | 0.4 | 204.75 | 2.04 | **0** |    | 0.4 | 290.13 | 3.45 | **0** |
|    | 0.6 | 407.99 | 0 | 0 |    | 0.6 | 522.72 | 0 | 0 |
|    | 0.8 | 577.35 | 0.98 | **0** |    | 0.8 | 773.15 | 0.38 | **0** |
|    | 1 | 817.92 | 0 | 0 |    | 1 | 1074.76 | 0.45 | **0** |
| 26 | 0.1 | 18.68 | 13.22 | **0** | 27 | 0.1 | 25.52 | 11.41 | **0** |
|    | 0.2 | 101.98 | 2.88 | **0** |    | 0.2 | 121.95 | 1.83 | **0** |
|    | 0.4 | 323.20 | 1.18 | **0** |    | 0.4 | 365.00 | 0.75 | **0** |
|    | 0.6 | 594.95 | 1.88 | **0** |    | 0.6 | 665.93 | 0.89 | **0** |
|    | 0.8 | 920.61 | 0.37 | **0** |    | 0.8 | 1011.61 | 0.37 | **0** |
|    | 1 | 1195.72 | 0.35 | **0** |    | 1 | 1327.83 | 0.28 | **0** |

**Notes.**
Best results are shown in bold.

we change the value within a range, while keeping the other parameter constant, and perform DDPBP to solve one random instance for each FLMP setting. In addition, we use the Friedman test to determine if there exist statistical differences among different parameter settings.

Table 5 indicates that the setting of $cn = 8$ leads to less time consumptions for all instances. For example, for FLMP with 24 activities and 0.9 density level, the solving times under 2 and 8 cores are 151 s and 68.39 s, respectively. In addition, the Friedman test shows that changing the number of applied cores leads to significant differences on algorithm performances with $p$-value of 2.29e−07, which confirms the necessity of utilizing all the computing resources for solving FLMP. Meanwhile, from Table 6, we observe that DDPBP performs marginally better when $na = 5$, and the $p$-values of varying $na$ is 0.92, which means that changing the workloads of forward and backward processes does not impact much the performance of the algorithm.

**Table 5  Solving time (seconds) under different *cn*.**

| n | den | cn | | | | n | den | cn | | | |
|---|---|---|---|---|---|---|---|---|---|---|---|
| | | 2 | 4 | 6 | 8 | | | 2 | 4 | 6 | 8 |
| 18 | 0.1 | 1.98 | 1.2 | 1.06 | **1** | 22 | 0.1 | 33.83 | 20.47 | 17.15 | **16.04** |
| | 0.5 | 1.8 | 1.12 | 1.38 | **0.93** | | 0.5 | 33.82 | 20.12 | 17.45 | **16.07** |
| | 0.9 | 1.8 | 1.12 | 1.07 | **1** | | 0.9 | 33.83 | 20.4 | 16.93 | **15.32** |
| 20 | 0.1 | 7.73 | 4.56 | 3.82 | **3.57** | 24 | 0.1 | 152.67 | 92.68 | 76.53 | **72.34** |
| | 0.5 | 7.6 | 4.86 | 4.08 | **4.19** | | 0.5 | 152.13 | 91.55 | 76.28 | **69.3** |
| | 0.9 | 7.56 | 4.99 | 3.95 | **3.66** | | 0.9 | 151 | 90.99 | 75.7 | **68.39** |

**Notes.**
Best results are shown in bold.

**Table 6  Solving time (seconds) under different *na*.**

| n | den | na | | | | | | |
|---|---|---|---|---|---|---|---|---|
| | | 3 | 5 | 7 | 9 | 11 | 13 | 15 |
| 18 | 0.1 | 1.27 | **1.18** | 1.3 | 1.67 | 1.22 | 1.44 | 1.75 |
| | 0.5 | 1.29 | 1.45 | **1.21** | 1.23 | 1.83 | 1.46 | 1.53 |
| | 0.9 | 1.52 | 1.8 | 1.59 | **1.35** | 1.36 | 1.4 | 1.91 |
| 20 | 0.1 | 5.62 | 5.54 | 5.57 | 5.87 | 4.87 | **4.75** | 4.8 |
| | 0.5 | 4.86 | 4.8 | 4.85 | 4.9 | **4.77** | 4.92 | 5.07 |
| | 0.9 | 4.95 | **4.89** | 4.93 | 5.51 | 5.6 | 5.52 | 5.51 |
| 22 | 0.1 | 22.35 | **21.77** | 23.94 | 22.16 | 21.94 | 21.77 | 21.79 |
| | 0.5 | 21.76 | 21.63 | **21.54** | 21.91 | 22 | 22.28 | 21.8 |
| | 0.9 | 21.87 | **21.66** | 22.34 | 21.78 | 21.79 | 22.11 | 22.02 |
| 24 | 0.1 | 100.87 | 102.68 | 102.44 | 101.59 | 103.44 | 101.08 | **100.32** |
| | 0.5 | 102.16 | 99.11 | 98.81 | 99.9 | 98.54 | **98.46** | 99.22 |
| | 0.9 | 99.35 | 98.78 | 99.13 | 99.15 | **98.47** | 100.28 | 101.59 |

**Notes.**
Best results are shown in bold.

## Strategy analysis

In order to confirm the validity of important strategies employed by the proposed algorithm, we produce three variants for comparisons, including: DDPBP-DDS that only uses the first decomposition, DDPBP-RCS without result-compression and DDPBP-HAS whose hash-address strategy related components have been removed. The addition experiments follow the same experimental protocol as Section 'Benchmark instances and experimental protocol'.

### *Effectiveness of double-decomposition strategy*

The double-decomposition strategy allows DDPBP to make full use of available computing resources. To evaluate the rationality, we create a variant DDPBP-DDS, where the second decomposition has been disabled. Hence, this variant only deploys the forward and backward processes on two cores to explore the search tree.

Table 7 presents the solving times obtain by two algorithms. The results show that DDPBP performs significantly better than its variant for all instances ($12 > CV^{12}_{0.05} \approx 9$).

Shang et al. (2023), *PeerJ Comput. Sci.*, DOI 10.7717/peerj-cs.1597

**Table 7   Solving time (seconds) obtained by DDPBP and DDPBP-DDS.**

| $n$ | $den$ | Solving time | | $n$ | $den$ | Solving time | |
|---|---|---|---|---|---|---|---|
| | | **DDPBP** | **DDPBP-DDS** | | | **DDPBP** | **DDPBP-DDS** |
| 18 | 0.1 | **1.25** | 2.84 | 20 | 0.1 | **3.79** | 12.98 |
| | 0.5 | **1.11** | 3.08 | | 0.5 | **3.65** | 13.69 |
| | 0.9 | **1.03** | 2.95 | | 0.9 | **3.68** | 12.55 |
| 22 | 0.1 | **16.83** | 61.77 | 24 | 0.1 | **87.16** | 316.72 |
| | 0.5 | **16.6** | 62.77 | | 0.5 | **78.1** | 260.15 |
| | 0.9 | **20.46** | 66.83 | | 0.9 | **68.04** | 227.57 |

**Notes.**
  Best results are shown in bold.

Specifically, the average gap of solving time ($avg(Variant - DDPBP)/Variant$) is 69.02%. To conclude, this experiment confirms that the proposed DDPBP algorithm is enhanced by the double-decomposition strategy.

### *Effectiveness of result-compression strategy*

The result-compression strategy is designed for reduce the communication cost when each core finishes tasks and transmits results. To assess the role of this strategy, we produce a variant DDPBP-RCS, where the cores send the resulting $SetT\{i\}$ directly, instead of transmitting extracted information. In Table 8, column "Row 3–9" shows the total amount of data transmission when two algorithms are about to end the explorations for rows 3, 7 and 9 of the search trees, and column "Time" presents the corresponding solving time for each instance. It should be noted that the density level does not affect the size of $SetT\{i\}$ or the amount of extracted information, hence the amount of data transmission maintains for instances with the same number of activities.

From Table 8, we observe that for all instances, DDPBP obtain the optimal solution with less time and transmission costs ($12 > CV_{0.05}^{12} \approx 9$). For example, for the instance with 22 activities and 0.5 density level, DDPBP spends 16.17 s to reach the optimum, and transfers 70.99MB data when finishing the exploration of Row 7, while the experimental results of the variant are 23.19 s and 239.46 MB. In general, the average gap of data amount and solving time are 62.46% and 22.18%, respectively. One reason for this result is that the result-compression strategy only delivers the key information of a sparse $SetT\{i\}$ within each core, which reduces the amount of data transmission, and thus improves the efficiency of the parallel framework.

### *Effectiveness of hash-address strategy*

The hash-address strategy is introduced to accelerate the process of locating similar nodes in $SetT\{i\}$ during the forward and backward processes. To evaluate the impact of this strategy, we create a variant DDPBP-HAS, which identifies whether two nodes are similar by comparing activities within nodes. Hence, in order to find similar nodes, the variant must check all the nodes stored in $SetT\{i\}$.

Table 9 shows that DDPBP significantly outperforms its variant for spending much less time on all instances ($12 > CV_{0.05}^{12} \approx 9$). For example, for the instance with 17 activities

**Table 8** Data transmission amount (MB) and solving time (seconds) obtained by DDPBP and DDPBP-RCS.

| n | den | DDPBP | | | | n | den | DDPBP-RCS | | | |
|---|-----|-------|---|---|------|---|-----|-----------|---|---|------|
|   |     | Row 5 | Row 7 | Row 9 | Time |   |     | Row 5 | Row 7 | Row 9 | Time |
|    | 0.1 |      |       |        | **1.3**  |    | 0.1 |       |        |         | 1.35 |
| 18 | 0.5 | **0.49** | **12.04** | **20.42** | **1.02** | 18 | 0.5 | 0.98 | 36.95 | 56.42 | 1.29 |
|    | 0.9 |      |       |        | **1.06** |    | 0.9 |       |        |         | 1.2  |
|    | 0.1 |      |       |        | **3.54** |    | 0.1 |       |        |         | 4.91 |
| 20 | 0.5 | **0.74** | **31.65** | **72.56** | **3.98** | 20 | 0.5 | 1.49 | 99.41 | 215.33 | 5.37 |
|    | 0.9 |      |       |        | **3.88** |    | 0.9 |       |        |         | 4.91 |
|    | 0.1 |      |       |        | **16.16** |   | 0.1 |       |        |         | 22.16 |
| 22 | 0.5 | **1.05** | **70.99** | **218.53** | **16.17** | 22 | 0.5 | 2.19 | 239.46 | 698.33 | 23.19 |
|    | 0.9 |      |       |        | **16.83** |   | 0.9 |       |        |         | 22.85 |
|    | 0.1 |      |       |        | **71.72** |   | 0.1 |       |        |         | 96.5 |
| 24 | 0.5 | **1.47** | **153.86** | **627.04** | **77.95** | 24 | 0.5 | 3.12 | 528.16 | 1995.14 | 94.26 |
|    | 0.9 |      |       |        | **68.16** |   | 0.9 |       |        |         | 95.25 |

Notes.
Best results are shown in bold.

**Table 9** Solving time (seconds) obtained by DDPBP and DDPBP-HAS.

| n | den | Solving time | | n | den | Solving time | |
|---|-----|-------|-----------|---|-----|-------|-----------|
|   |     | DDPBP | DDPBP-HAS |   |     | DDPBP | DDPBP-HAS |
|    | 0.1 | **0.38** | 11.77 |    | 0.1 | **0.48** | 34.5 |
| 14 | 0.5 | **0.32** | 11.24 | 15 | 0.5 | **0.39** | 33.89 |
|    | 0.9 | **0.57** | 11.24 |    | 0.9 | **0.32** | 32.8 |
|    | 0.1 | **0.47** | 118.24 |   | 0.1 | **0.71** | 460.71 |
| 16 | 0.5 | **0.82** | 123.97 | 17 | 0.5 | **0.72** | 476.33 |
|    | 0.9 | **0.46** | 116.05 |   | 0.9 | **0.91** | 464.87 |

Notes.
Best results are shown in bold.

and 0.5 density level, the solving times of DDPBP and its variant are 0.72 s and 476.33 s respectively. In general, the average gap of solving time is 98.61%, and as the number of activities increases, the solving times of the variant increase rapidly. This experiment proves the necessity of hash-address strategy for the proposed algorithm.

## CONCLUSIONS

Minimizing the total feedback length is an effective objective to optimize development projects. In this study, we presented an efficient double-decomposition based parallel branch-and-prune algorithm, to obtain the optimal activity sequence of FLMP. The proposed algorithm divides FLMP into several subproblems through an original double-decomposition strategy, then employs multiple CPU cores to solve them concurrently. In addition, we proposed a result-compression strategy to reduce communication costs in parallel process, and a hash-address strategy to boost the efficiency of sequence comparisons.

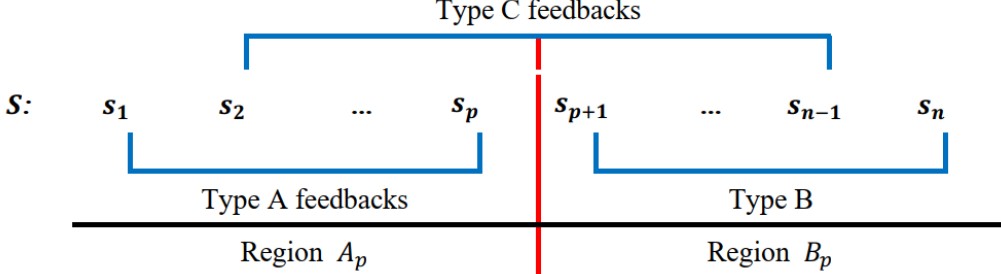

**Figure A1  Three feedback types.**

Computational experiments indicate that the proposed algorithm is able to increase the scale of FLMP that exact algorithms can solve within 1 h to 27 activities, and clearly outperforms the best exact algorithms in literature. Furthermore, additional experiments show the effects of two parameters on algorithm performances, and confirm the advantage of the double-decomposition strategy, the importance of the result-compression strategy and the hash-address strategy.

Some strategies applied in this study are general and could be introduced to solve other sorting problems. For example, the double-decomposition strategy first divides a sorting problem into forward and backward subproblems, then further decomposes them into several sorting tasks, which can significantly reduce the complexity of sorting problems. Furthermore, the hash-address strategy maps similar sequences into a unique value, which can be used to compare and search sequences.

## ACKNOWLEDGEMENTS

We are grateful to the PeerJ editors and anonymous reviewers for their helpful comments and suggestions.

## APPENDIX

As shown in Fig. A1, when sequence $S = (s_1, s_2, \ldots, s_p, s_{p+1}, \ldots, s_n)$ is split by position $p$, all feedbacks are divided into three types.

**Type A:** Feedbacks between activities in region $A_p$, such as the feedback from activity $s_p$ to $s_1$. The total feedback length $fl_p^a$ is as follows:

$$fl_p^a = \sum_{h=1}^{p-1} \sum_{k=h+1}^{p} d_{s_h s_k}(k-h) \tag{16}$$

Equation(16) is the first item of Eq. (7), which means that $fl_p^a$ is a part of feedback value $fv_p^a$ in $A_p$. Since $fl_p^a$ is only related to $A_p$, changing the subsequence in $B_p$ can not affect its value.

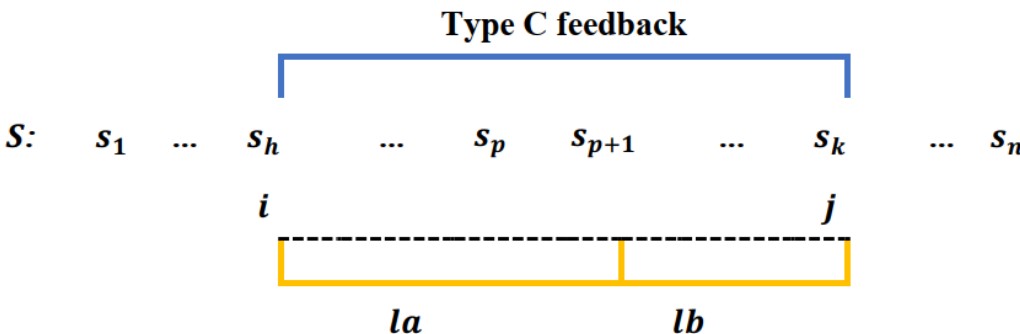

**Figure A2** **Further decomposition of type C feedbacks.**

**Type B:** Feedbacks between activities in region $B_p$, such as the feedback from activity $s_n$ to $s_{p+1}$. The total feedback length $fl_p^b$ is as follows:

$$fl_p^b = \sum_{h=p+1}^{n-1} \sum_{k=h+1}^{n} d_{s_h s_k}(k-h) \tag{17}$$

Equation (17) is the first item of Eq. (8), which means that $fl_p^b$ is a part of feedback value $fv_p^b$ in $B_p$. Since $fl_p^b$ is only related to $B_p$, changing the subsequence in $A_p$ can not affect its value.

**Type C:** Feedbacks from region $B_p$ to $A_p$, such as the feedback from activity $s_{n-1}$ to $s_2$. The total feedback length $fl_p^c$ is as follows:

$$fl_p^c = \sum_{h=1}^{p} \sum_{k=p+1}^{n} d_{s_h s_k}(k-h) \tag{18}$$

Since type C feedback spans two regions, changing subsequences in $A_p$ or $B_p$ can affect $fl_p^c$, which means that this type of feedback is not independent of any regions. Apparently, the total feedback length of FLMP consists of three types of feedback length, i.e., $fl = fl_p^a + fl_p^b + fl_p^c$.

In Fig. A2, assume that subsequences $SA_p$ and $SB_p$ are fixed. Without loss of any generality, set activities $s_h = i$ and $s_k = j$, hence the feedback between activities $i$ and $j$ is type C, and its length $l = k - h$. Further, we divide $l$ into $la = (p+1) - h$ and $lb = k - (p+1)$.

If we fix activity $i$ at position $h$, and move activity $j$ to any position in region $B_p$, $la$ remains unchanged, which means that $la$ is not affected by the subsequence in $B_p$. The same is true for $lb$. Then, we divide $fl_p^c$ into feedback values $fv_p^{ca}$ and $fv_p^{cb}$, which are only related to $A_p$ and $B_p$, respectively.

$$fv_p^{ca} = \sum_{h=1}^{p} \sum_{k=p+1}^{n} d_{s_h s_k}(p+1-h) \tag{19}$$

$$fv_p^{cb} = \sum_{h=1}^{p} \sum_{k=p+1}^{n} d_{s_h s_k}(k-p-1) \tag{20}$$

Eqs. (19) and (20) are the second items of Eqs. (7) and (8). We have $fl_p^c = fv_p^{ca} + fv_p^{cb}$, and the following equation:

$$fl = fl_p^a + fl_p^b + fl_p^c = (fl_p^a + fv_p^{ca}) + (fl_p^b + fv_p^{cb}) = fv_p^a + fv_p^b \tag{21}$$

According to Eq. (21), $fv_p^a = fl_p^a + fv_p^{ca}$ and $fv_p^b = fl_p^b + fv_p^{cb}$ are only related to $A_p$ and $B_p$, respectively. In other words, Property 1 and 2 hold.

### Funding
This work was supported by the Natural Science Basic Research Program of Shaanxi (No. 2023-JC-QN-0793, 2022JM-423), the Special Foundation for Philosophy and Social Science Research of Shaanxi (No. 2023QN0036), Scientific Research Plan Project of Shaanxi Provincial Department of Education (21JP007), the Fundamental Research Funds for the Central Universities (No. 300102231656, No. 300102233612). The funders had no role in study design, data collection and analysis, decision to publish, or preparation of the manuscript.

### Grant Disclosures
The following grant information was disclosed by the authors:
The Natural Science Basic Research Program of Shaanxi: No. 2023-JC-QN-0793, 2022JM-423.
The Special Foundation for Philosophy and Social Science Research of Shaanxi: No. 2023QN0036.
Scientific Research Plan Project of Shaanxi Provincial Department of Education: 21JP007.
The Fundamental Research Funds for the Central Universities: No. 300102231656, No. 300102233612.

### Competing Interests
Jin-Kao Hao is an Academic Editor for PeerJ.

### Author Contributions

- Zhen Shang conceived and designed the experiments, performed the experiments, analyzed the data, performed the computation work, prepared figures and/or tables, and approved the final draft.
- Jin-Kao Hao conceived and designed the experiments, performed the computation work, authored or reviewed drafts of the article, and approved the final draft.
- Fei Ma conceived and designed the experiments, performed the experiments, analyzed the data, authored or reviewed drafts of the article, and approved the final draft.

### Data Availability
The FLMP instances and the corresponding optimal solutions, and the DDPBP source code are available on GitHub and Zenodo:

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
