# Peer review of "A double-decomposition based parallel exact algorithm for the feedback length minimization problem"

_PeerJ Computer Science, doi:10.7717/peerj-cs.1597_

## Round 0.1 · original submission · Major Revisions

I share the opinion of the two reviewers that the paper is very interesting. However both raised comments that require to be addressed before the publication.
In particular provide the requested comparison with respect to recent related works.

Reviewer 1 ·

Basic reporting

This research paper introduces a double-decomposition-based parallel branch-and-prune algorithm for scheduling interrelated activities. The proposed algorithm aims to minimize the total feedback length (FLMP) by decomposing the problem into subproblems and utilizing available computing resources concurrently. To further enhance its performance, the algorithm incorporates a result-compression strategy and a hash-address strategy.

Following are some of the suggestions for improvement that must be addressed for the acceptance of the paper:
- In the introduction section, the paper presented 1 recent study relate to the presented problem i.e., Khanmirza et al. (2021) [Page 3, line 74]. The authors should discuss more research approaches that were proposed recently i.e., in the Years 2021, 2022, and 2023. Too much discussion on classical approaches is not suitable.
- As mentioned in the above point, the authors must discuss newer approaches (The year 2021 to 2023) and discuss the research gap and motivate their study, discussion as compared to the classical approach is not sufficient w.r.t to the novelty of the work.
- Overhead and complexity analysis of the proposed approach should be provided along with the other mentioned approaches (state-of-the-art techniques of the Year 2021 onwards).
- The introduction section should clearly mention the core research contributions and novelty (considering the state-of-the-art approaches, not the classical ones).
- There is no literature review Section. There is a very direct highlight of the existing approaches in the introduction Section. Authors must provide a separate Section of the literature review and discuss technique by technique. Moreover, a comparison Table (related work summary) should be provided with research gaps/shortcomings of the existing approaches.
- The proposed algorithm and its working mechanism have been discussed in detail and comprehensively; however, more discussion on the complexity analysis is needed.
- The evaluation of the approach should consider newer techniques i.e., 2021 or onward research approaches.

Experimental design

-Experimental setup should be defined separately under a sub-section.
- The evaluation of the approach should consider newer techniques i.e., 2021 or onward research approaches.

Validity of the findings

-

Additional comments

The novelty of the approach should be justified with respect to the state-of-the-art approaches (2021 onwards proposed techniques).

Reviewer 2 ·

Basic reporting

The authors of this paper tackled the "Feedback length minimization problem", which consists of scheduling interrelated activities in an appropriate sequence, when developing software products. The problem is interesting, and it is based on a real application.

The authors propose a parallel branch-and-prune algorithm based on double decomposition to tackle the problem. The approach is interesting and has been shown to be useful for this problem, helping to identify new optimal solutions.

The paper is well-written, and it is easy to follow. However, the authors should consider some concerns that are explained next, before the paper could be accepted for publication.

Experimental design

- If I am right, every row in Table 1 represents the average running time of each algorithm for 10 instances. Is that right? Could you please explicitly indicate the total number of instances in Section 5.1. Furthermore, the explanation of the parameters used (i.e., activities and density) is set in section 5.2. Please move it to Section 5.1 (this is where a reader expect that information).

- In Table 1, you report just average time to solve the instances, however, I bet that for those instances where the algorithm was not able to find a solution, they found a feasible solution. Could you provide the gap reported by the methods?

- I know you are proposing an exact method and that the method is compared with other exact methods but, could you provide some insight about the performance of the state-of-the-art heuristic algorithms for the tackled instances? Particularly, what is the performance of heuristic approaches for the instances where the optimum is found?

- I found the source code and the DSM generator as supplementary material for review. However, it is unclear to me if this is going to be publicly available. Could you explain if this is happening and how? Furthermore, could you make it publicly available the specific instances used in your experiments?

- Could you provide the optimal value found per instance in any public URL? or in an Appendix?

- How does the number of available cores affect your proposal? Have you tried the performance of the method when reducing the number of available cores?

Validity of the findings

Experimental results indicate that the proposed algorithm finds the optimal sequence for FLMP in 27 activities within 1 hour. Additionally, it outperforms previous state of the art exact algorithms in terms of computing time and size of the instance solved.

The proposed approach is interesting. The exploitation of multiple cores collaborating together to enhance the performance of the algorithm results very interesting, and it can be applied to other related problems.

The hashing strategy is also interesting for other exact methods where the number of solutions to be explored is very large.

Additional comments

- Page 1. Line 38. Please reword the sentence "And other information..."
- In Fig 1.a and Fig 1.b, please change the "......" with just a "...".
- Page 2. Line 56, please use \noindent at the beginning of the sentence
- Page 3. Line 66, please use \noindent at the beginning of the sentence
- Page 3. Line 68, please reword "multi positions" into "multiple positions"
- Fig. 2. Please improve the quality of the figure. I recommend the use of vector image formats such as PDF, SVG or EPS
- Fig. 3. Please improve the quality of the figure. I recommend the use of vector image formats such as PDF, SVG or EPS
- Fig. 4. Please improve the quality of the figure. I recommend the use of vector image formats such as PDF, SVG or EPS
- Fig. 5. Please improve the quality of the figure. I recommend the use of vector image formats such as PDF, SVG or EPS
- Page 9. Line 244, please use \noindent at the beginning of the sentence
- Page 15. Section 7. Please reword the title of the section into "Conclusions"

---

## Round 0.2 · accepted · Accept

I would suggest addressing the final, minor comments, of reviewer 2.

Reviewer 1 ·

Basic reporting

The revisions are satisfactory and there are no further comments from my side.

Experimental design

The revisions are satisfactory and there are no further comments from my side.

Validity of the findings

The revisions are satisfactory and there are no further comments from my side.

Additional comments

The revisions are satisfactory and there are no further comments from my side.

Reviewer 2 ·

Basic reporting

In my previous report I already indicated that the problem addressed in this paper was interesting, and it based on a real application. The authors proposed a novel approach which is interesting and has been shown to be useful for this problem, helping to identify new optimal solutions. The paper is well-written, and it is easy to follow.

Furthermore, I suggested a detailed list of concerns and all of them have been successfully addressed by the authors, in the reviewed version of this paper.

Experimental design

My previous suggestions in this regard have been successfully addressed by the authors, in the reviewed version of the paper. I found the table with the gap values very interesting. The comparison with previous heuristics is also useful and provides a clearer idea about the performance of the methods proposed.

I only add two new minor concerns:
- Tables 3 and 4 do not use bold type font to highlight the best results, which does not follow the format of the rest of the tables.
- Also, I think it might be useful to include in some of the tables a row with the average value obtained by each method. Maybe one Avg. row per "n"? Does it make sense to have an Avg. for the whole table? This is just a suggestion, I let you decide whether include it or not.

Validity of the findings

I have nothing to add with respect to my previous comments in this regard.

Additional comments

My previous suggestions in this regard have been successfully addressed by the authors, in the reviewed version of the paper.